# Automatic Segmentation in Abdominal CT Imaging for the KiTS21 Challenge

Jimin Heo

`gjwlals111@gmail.com`

**Abstract.** With the KiTS21 Grand Challenge, I propose the automatic segmentation model between the kidney and the mass of the kidney which includes tumor and cyst. Convolutional Neural Network is trained in patches of three-dimensional abdominal CT imaging. For the segmentation of the 3D image, a variant of U-Net which consists of 3D Encoder-Decoder CNN architecture with additional Skip Connection is used. Lastly, there is a loss function to resolve the class imbalance problem frequently occurring in the task of medical imaging. Sørensen-Dice Score and Surface Dice Score on the validation are 82.52 and 70.45.

**Keywords:** KiTS21 Challenge · 3D Encoder-Decoder U-Net · Medical Imaging

## 1   Introduction

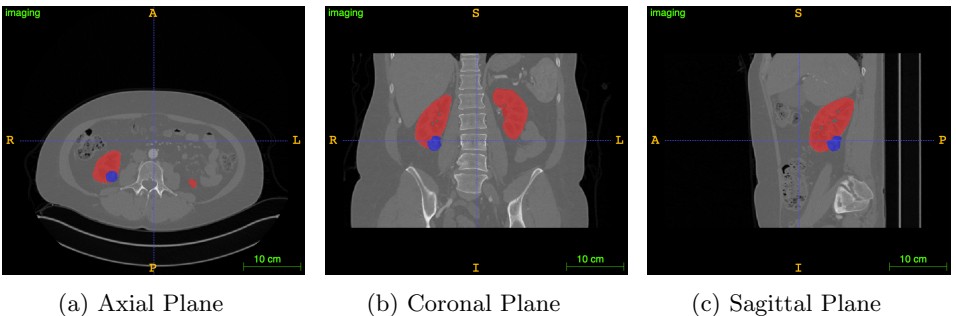

(a) Axial Plane          (b) Coronal Plane          (c) Sagittal Plane

Fig. 1: Examples of KiTS21 challenge dataset. Kidney class is shown in red, tumor class is shown in blue. Not shown in the figure, but cyst class is in green.

According to Association of American Medical College, a shortage of between 40,800 and 104,900 physicians by 2030 will occur in the United States. Moreover, as reported by International Agency for Research on Cancer, more than 400,000 people are affected by kidney cancer in each year and resulted in 175,000 deaths globally. Kidney is involved in removing wastes and excretion of metabolites

in the body, and is also responsible for important functions such as moisture and electrolyte balance, acid-alkaline maintenance, and control of other organ functions by producing hormones and vitamins, which may affect surrounding tissues or organs. In addition, tumors in the kidneys are often found after lesions are transmitted to other organs with no special awareness at first, and are often found during tests for other internal diseases due to various non-specific symptoms and signs. The most precise way to evaluate tumors occurring in the kidneys is to take abdominal CT images, so it is important to analyze the acquired images quickly to extract kidneys and tumors. As a way to automate this, the paper presents a method for segmenting abdominal CT images into kidneys, tumors and cysts for KiTS21 which is the grand challenge of Kidney and Tumor Segmentation in 2021 via Deep Convolutional Neural Network model.

## 2 Methods

Based on U-Net architecture, the winner[3] on KiTS19 developed the variants of the architecture. In the methods, motivated by the winner's model, I propose the model which is the variant of the winner's. With the open source framework MIScnn[5], methods for KiTS21 challenge are implemented in Keras using the Tensorflow backend.

### 2.1 Training and Validation Data

Our submission made use of the official KiTS21 training set alone. For target data, only voxel-wise majority voting is used.

### 2.2 Preprocessing

In the 3D Volumetric Image processing, all large dataset cases such as the KiTS dataset should be resampled in common spacing. The reason is that, though the voxel spacings of the cases are generally inconsistent, deep learning neural networks cannot interpret voxel spacings. In order to input the data without that problem, all of the voxel spacings become 3.22 x 1.62 x 1.62 mm. After resampling, because the range of HU(Hounsfield Unit) values is too large to train, I clip the value to [-79, 304] which is the range of fat to soft tissue. Furthermore, Normalization is used following clipping to limit the range and set the standard distribution. Succeeding those preprocessing for the volume, data augmentation is required to regularize overfitting. The method consists of linear and non-linear transformations such as scaling, rotating, symmetric and elastic deformation[7]. Also, Gaussian noise, intensity and contrast algorithms such as gamma correction are included. Lastly, before training, patchwise-crop which analysis of random cropped patches by 80 x 160 x 160 from the image is performed.

### 2.3   Proposed Method

Although the challenge evaluates only performance, in terms of light-weight and acceleration of both training and inference time, I describe the one-stage semantic segmentation model.

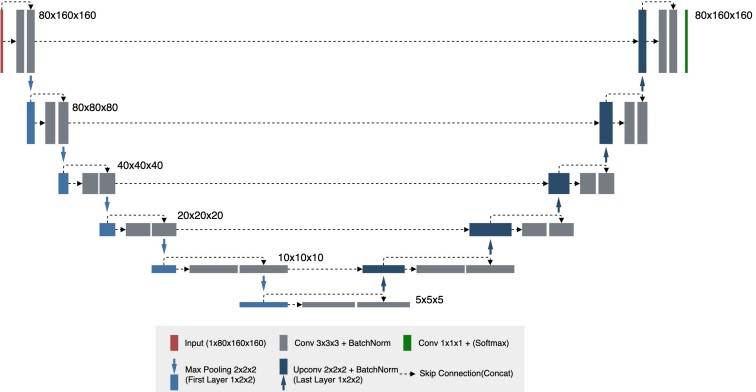

Fig. 2: Network Architecture based on 3D Encoder-Decoder U-Net

**Network Architecture** It is similar to the winner's model. Basically, both shapes of this model and the winner's model are from 3D U-Net Encoder-Decoder architecture[6]. First, to make a difference and achieve better performance, additional Skip Connections are used. Skip Connection can be split up two ways. One is Residual Connection[1] that is sum of layers. The other is Dense Connection[2] that is concatenation of layers. To maximize the propagation of information, Dense Connection is chosen. Second, Max-pooling is included in the network architecture to extract the maximum value of the feature map. Also, there are Activation Function such as ReLU and Batch Normalization in Transposed Convolution.

**Loss Function** In Object Detection or Segmentation, especially in medical tasks such as KiTS21, class imbalance is one of the most important issues. As mentioned, in the challenge, the kidney mass class and even kidney class are much smaller than the background class. To overcome class imbalance, there is Focal loss[4] function which is a variant of Cross-Entropy loss. To use with Softmax, $\alpha$-factor is modified from scalar to vector. And, there is gamma-factor from the best result in the experiment of Lin et al[4]. Also, Dice Loss is utilized to perform better. In conclusion, the architecture uses sum of Focal Loss and Dice Loss for loss function.

**Optimization and Validation Strategy** For optimization, Adam Optimizer with lr=3e-4 is chosen. Additionally, monitoring the validation metric on each epoch, a strategy to prevent underfitting is used. The strategy is reducing the learning rate on plateau. If there is no room for improvement in the metric of the validation set, then the scheduler reduces learning rates to induce improvement of the metric. In the paper, the scale factor is 0.1 and patience is 150. The minimum threshold is 1e-4 and the minimum limitation of the learning rate is 3e-6.

## 3   Results

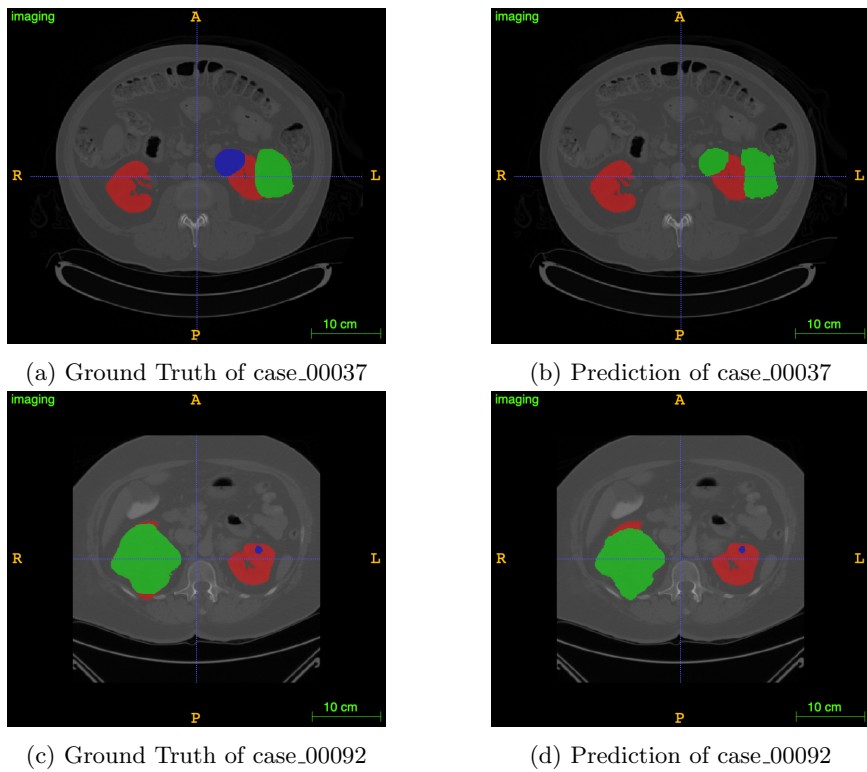

(a) Ground Truth of case_00037             (b) Prediction of case_00037

(c) Ground Truth of case_00092             (d) Prediction of case_00092

Fig. 3: Examples of Ground Truth and Prediction. Kidney class is shown in red, tumor class is shown in blue. cyst class is in green. There is case_00037 predicted incorrectly in the tumor as the cyst. Whereas, case_00092 is predicted more accurately than case_00037.

Totally, 300 cases are released publicly for the challenge. In the experiment, randomly 240 cases are selected for training and 60 cases are selected for val-

idation. I train the model on local GPU, NVIDIA RTX 3090 24GB. Training costs about three days. With batch size of 2 and 500 epochs, The results of the validation score are below.

| Kidney Dice | Mass Dice | Tumor Dice | **Mean Dice** |
|---|---|---|---|
| 93.69 | 78.83 | 75.03 | **82.52** |
| Kidney SD | Mass SD | Tumor SD | **Mean SD** |
| 87.13 | 64.05 | 60.16 | **70.45** |

Table 1: Sørensen-Dice Score and Surface Dice Score on the validation

## 4    Discussion and Conclusion

I described a one-stage semantic segmentation model for KiTS21 Challenge from 3D Abdominal CT imaging. With the model based on U-Net and the sum of Focal Loss and Dice Loss, I attempted to overcome Class Imbalance. Sørensen-Dice Score and Surface Dice Score on the validation are 82.52 and 70.45. For the better performance, some modules such as Atrous Spatial Pyramid Pooling or U-Net++ could be used.

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
