# OpenReview forum: "Automatic Segmentation in Abdominal CT Imaging for the KiTS21 Challenge"
_MICCAI.org/2021/Challenge/KiTS — Submitted to KiTS21 Challenge_

### Official Review · Reviewer_gHis · 2021-08-30

**Rating:** 8

**Review:**

This paper does a good job covering all details that were requested in the template, and the authors make effective use of figures and tables to organize their presentation and support their arguments. It might be nice to rephrase the paper title to be "Automatic Segmentation in Abdominal CT Imaging **for the** KiTS21 Challenge" rather than "... **with** KiTS21...". The authors should also be sure to include the final results on the test set once they become available.

---

### Official Review · Reviewer_BA1Y · 2021-08-30

**Rating:** 7

**Review:**

### Overall

- It is preferable to add an institutional email address rather than gmail, if possible
- It would be nice to add final results to the abstract and results section once known

### Introduction

- Looks good

### Methods

- Looks good

### Results

- In figure 3, it woudl be great if you could give the ids of the cases that you are showing

### Discussion and Conclusion

- Looks good

---

### Decision · Program_Chairs · 2021-08-30

**Decision:**

Minor Revisions

**Comment:**

Please address the reviewer comments and resubmit